# Adherence to Mediterranean Diet Related with Physical Fitness and Physical Activity in Schoolchildren Aged 6–13

**DOI:** 10.3390/nu12020567

**Published:** 2020-02-22

**Authors:** José Francisco López-Gil, Javier Brazo-Sayavera, Antonio García-Hermoso, Juan Luis Yuste Lucas

**Affiliations:** 1Departamento de Actividad Física y Deporte, Facultad de Ciencias del Deporte, Universidad de Murcia, 30720 San Javier, Region of Murcia, Spain; 2Polo de Desarrollo Universitario EFISAL, Centro Universitario Regional Noreste, Universidad de la República, 40000 Rivera, Uruguay; jbsayavera@cur.edu.uy; 3Navarrabiomed, Complejo Hospitalario de Navarra (CHN), Universidad Pública de Navarra (UPNA), IdiSNA, 31008 Pamplona, Navarra, Spain; antonio.garciah@unavarra.es; 4Laboratorio de Ciencias de la Actividad Física, el Deporte y la Salud, Universidad de Santiago de Chile, USACH, 71783-5 Santiago, Chile; 5Departamento de Expresión Plástica, Musical y Dinámica, Facultad de Educación, Universidad de Murcia (UM), 30100 Espinardo, Region of Murcia, Spain; jlyuste@um.es

**Keywords:** cardiorespiratory fitness, muscle strength, feeding patterns, lifestyle, children

## Abstract

The relationship between adherence to the Mediterranean Diet (MD) and both physical fitness (PF) and physical activity (PA) level has been analysed in several studies. The aim of this research was to describe, compare and analyse the level of PF and PA in schoolchildren aged 6–13 in the Region of Murcia, according to adherence to the MD. A descriptive and cross-sectional study was performed. A total of 370 schoolchildren (44.9% girls) aged 6–13 (8.7 ± 1.8) from six primary schools in the Region of Murcia (Spain). Mediterranean Diet Quality Index for children and teenagers (KIDMED) was used to determinate the adherence to the MD. The ALPHA-FIT Test Battery was applied for assess body composition and PF. PA level was determined using Physical Activity Questionnaire for Older Children (PAQ-C). Only 25.9% of the schoolchildren had optimal adherence to the MD. Regarding the scores of the different PF tests in MD groups, only statistically significant differences were found for cardiorespiratory fitness (CRF) (*p* = 0.048) in boys. PA level showed statistically significant differences in both boys (*p* = 0.040) and girls (*p* = 0.016). A positive relationship was found between the KIDMED and PA level (*p* = 0.235). A higher probability of having a greater CRF (OR = 1.17; CI = 1.02–1.34) and PA level (OR = 7.84; CI = 2.84–21.60) was found in high MD group. These results suggest that an optimal adherence to the MD is associated with higher CRF and PA level in the selected schoolchildren.

## 1. Introduction

Health-related physical fitness (PF) is defined as a set of attributes that people have or achieve related to the ability to engage in physical activity (PA) [1]. The components of PF shown to be directly related to improve health are cardiorespiratory fitness (CRF), flexibility, muscular fitness (i.e., this definition incorporates the domains of muscular strength, muscular power and local muscular endurance) and body composition [2]. Likewise, the association between PF and some metabolic risk factors in children, adolescents, and adults has been well described in previous researches [3,4]. Therefore, the development of PF and an adequate PA level are essential resources for maintaining vital functions; acting as a powerful predictor of morbidity and mortality throughout life [5,6].

There are several factors that influence the PF, such as genetics [7], biological characteristics [8] and lifestyle [9]. Thus, regarding lifestyle, inadequate eating habits are also important with respect to the onset and progression of metabolic disorders, playing an important role in the development and progression of cardiovascular diseases [10], and causing more deaths than other factors traditionally considered as smoking [11].

In terms of eating habits, Mediterranean Diet (MD) is postulated as one of the healthiest eating patterns that exist [12] due to its particular characteristics, such as high consumption of food of vegetable origin, olive oil, a certain quantity of dairy products, as well as an active lifestyle [13]. However, adherence to MD in the Mediterranean region has worsened over the years, especially among the children [14]. This could lead to a decline in their health, as well as an increased risk of cardiovascular disease, metabolic syndrome, overweight or obesity [15,16].

Childhood and adolescence are key stages in the acquisition of lifestyles, since a physically active life, in which people can acquire an adequate level of PF, and healthy eating habits, are important determinants of present and future health [5,17]. For this reason, the relationship between adherence to the MD and different parameters of PF has been analysed in several studies [18,19,20,21,22], as well as with PA level [18,19,22]. Thus, some studies point out that a combination of high levels of muscle strength and an optimal adherence to MD are associated with healthier metabolic profile [20,21], as well as to a better health-related quality of life [23]. In addition to this, a positive association has been found between adherence to the MD and CRF [18,19,23,24], as well as speed-agility [25], but only in males. Likewise, some studies found positive associations between higher PA level (assessed by the Physical Activity Questionnaire for Older Children (PAQ-C) and greater adherence to the MD [18,19,22]. This fact could be explained by lifestyle factors, which could interact with each other in a synergistic way to influence PA level [25,26].

Notwithstanding, some of the positive associations cited above were found in studies developed outside the Mediterranean region like South America [19,20,21], where the type of food could vary with those found in the Mediterranean region [27]. Moreover, the differences among continents on food environment, defined as all aspects of the local environment that influence diet (i.e., community, organizational and consumer food environment), could have influence on the eating patterns [28]. Thus, the Region of Murcia is located in the south of Spain, where several studies have been carried out [22,29]. However, most of these studies have focused on adolescents and only few of them have been conducted on primary schoolchildren. Thus, we found a gap in the scientific literature regarding the evaluation of the associations among MD, PF and PA at this age stage. In addition, the report card on PA indicators from the Global Matrix 3.0 reported insufficient information about PF in children (and adolescents) in Spain [30], therefore there is a need of information about this indicator due to it decreasing trend [31].

Based on the evidence and the lack of information in different regions of Spain, the aim of this research was to describe, compare and analyse the level of PF and PA in schoolchildren aged 6 to 13 in the Region of Murcia, according to adherence to the MD.

## 2. Materials and Methods

### 2.1. Design and Participants

A descriptive and cross-sectional study was performed. A total of 370 schoolchildren (204 boys and 166 girls) aged 6–13 (8.7 ± 1.8) from six primary schools in the Region of Murcia (Spain) with similar sociodemographic characteristics participated in the study. For this purpose, the sample was selected using a nonprobability sampling. However, although we used this type of sampling, all schoolchildren from the selected schools were invited to participate.

For participation in the study, we included all children whose parents/legal guardians authorised the inclusion of their children or guardians. These parents and their children were previously informed about the purpose of the study and the nature of the tests that would be performed, through an informative document. As an exclusion criterion, we did not include those who were exempt from participation in Physical Education. Both the PF tests and the completion of the questionnaires were carried out by the schoolchildren during the physical education sessions.

This research was conducted in accordance with the Helsinki Declaration and with full respect for the human rights of the study participants and was approved by the Bioethics Committee of the University of Murcia (ID 2218/2018).

### 2.2. Procedures

#### 2.2.1. Adherence to the Mediterranean Diet (MD)

##### Mediterranean Diet Quality Index for Children and Teenagers (KIDMED Index)

In order to determine the adherence to the MD, the KIDMED index was used [32]. This is a test widely used in Spanish children and teenagers [18,29], which was created and validated by Serra-Majem et al. [33]. The index varies from 0 to 12 and is based on a 16-question test. Questions that present negative aspects in relation to the MD are scored with a value of −1, and those with positive aspects with +1. The sum of all values from the administered test is categorised into three different levels: (1) >8, optimal MD; (2) 4–7, improvement needed to adjust intake to Mediterranean patterns; (3) ≤3, very low diet quality [32].

#### 2.2.2. ALPHA-FIT Test Battery

PF tests included in the ALPHA-FIT Test Battery for children and adolescents [34] were used to assess the components that are described as follows.

##### Anthropometric Measurements

The height of the participants was determined using a portable height rod with an accuracy of 0.1 cm (Leicester Tanita HR 001, Tokyo, Japan). The body weight of the subjects was measured using an electronic scale (with an accuracy of 0.1 kg) (Tanita BC-545, Tokyo, Japan). The body mass index (BMI) was calculated from the ratio between body weight (kg) and the height squared of the participants (m^2^). Moreover, BMI z-score was determined using the World Health Organization (WHO) age-specific and sex-specific thresholds [35]. Waist circumference was measured to the nearest 0.1 cm at the level of the umbilicus, using a constant tension tape. Skinfold measurements to the nearest 0.2 mm were taken with calibrated steel callipers (Holtain, Crosswell, Crymych, UK) at the triceps, biceps, subscapular and iliac crest. These procedures were taken in accordance with the recommendations of the International Society for the Advancement of Kinanthropometry (ISAK). The log of the sum of skinfolds was used to calculate body density [36]. The Siri formula was used to calculate body fat from body density [37] and fat-free mass was estimated as the difference between total body mass and fat mass.

##### Cardiorespiratory Fitness (CRF)

The maximum volume of oxygen consumed was estimated by performing a maximum incremental field test (20 m Shuttle Run Test). Participants were tested to run between two lines 20 m apart while keeping a pace with the acoustic signals from a speakerphone audio player with Bluetooth technology. The initial speed was 8.50 km/h and was increased by 0.5 km/h every minute, reaching 18.0 km/h in the 20th minute. Participants were instructed to run in a straight line to pivot on completing the itinerary between the 2 lines and to follow the pace set by the audio signals. Subjects were encouraged to continue running if they were able to during the course of the test. The test ended when the participant failed to reach the end of the lines concurrent with the audio signals on 2 consecutive occasions. Otherwise, the test ended when the subject stopped because of fatigue. These stages were transformed to relative values of maximum oxygen consumption using Léger’s et al. [38] equations.

##### Muscular Fitness

Upper body muscular strength was evaluated by means of handgrip strength using a hand dynamometer with adjustable grip (TKK 5401 Grip D; Takei, Tokyo, Japan). Children were given a short demonstration and verbal order for the test and the dynamometer was regulated according to the child’s hand size as recommended previously [39]. The test was done in the standing position with the wrist in the neutral position and the elbow extended; children were given verbal support to “squeeze as hard as possible” and apply maximal strength for at least two seconds. Two attempts per hand were performed, and the best score was used. The average of the best scores achieved by each hand was used in the analysis [34]. Moreover, normalized handgrip strength was calculated as the average of the left and right and then expressed per kilogram of body weight [34].

Besides, lower body muscular strength was calculated by means of the standing broad jump. The participant stood behind the starting line, with feet together, and pushed off vigorously and jumped forward as far as possible. The distance was measured from the take-off line to the position where the back of the heel nearest to the take-off line lands on the floor. The test was done twice, and the best score was preserved (in cm) [34].

##### Motor Fitness

Speed-agility was measured by the 4 × 10 m Shuttle Run Test. Two lines, at a distance of 10 m, and two cones drawn were placed at the distant line. The participants ran as fast as possible from the starting line. Every time the participant crossed any of the lines, they picked up (the first time) or exchanged (second and third time) a sponge, which was previously placed behind the lines. The test was finished when the participant crossed the end line with one foot. Two attempts were performed, and the best score was retained (in seconds) [34].

#### 2.2.3. Physical Activity

##### Physical Activity Questionnaire for Older Children (PAQ-C)

Participants completed the PAQ-C to provide an estimate of the moderate-to-vigorous physical activity they engaged in. PAQ-C is a seven-day recall composed of nine items about the frequency of physical activities at school, at home, and during leisure time [40]. It contains nine items thatare rated on a five-point scale and has been validated and adapted to Spanish [41]. PAQ-C was validated for children aged 8–14. For this reason, in the case of children aged 6–7, the indications of Bervoets et al. [42] were followed, as it recommended that parents should be encouraged to support their children with reading and to complete the questions, provided that they did not give any guidance in answering the questions.

### 2.3. Statistical Analysis 

Means (M) and standard deviation (SD) are reported for all quantitative variables, and frequencies and percentages (%) are presented for all qualitative variables. Data normality was verified by a Kolmogorov–Smirnov test with Lilliefors correction, as well as the homogeneity of variances by a Levene test.

Subsequently, the data were analysed using Student’s *t*-test or Mann–Whitney *U* test for two-groups comparisons, and Kruskal–Wallis *H* test or one-way ANOVA for three group comparisons, depending on the compliance with the normality assumption. When differences between groups were observed, the post-hoc test were performed by Mann–Whitney U test with Bonferroni correction to account for the inflation of type-I error due to multiple comparisons made. Effect size was calculated by Cohen’s *d* (0.20, small; 0.50, medium; and 0.8, large effect). Associations between qualitative variables were determined using Pearson’s chi-square test. In addition, multinomial logistic regression was carried out in order to predict the probability of obtaining different results depending on the MD adherence category.

The relationships between quantitative variables were also determined using Spearman’s rho (*p*). Data analysis was performed using the software SPSS (IBM Corp, Armonk, NY, USA) for Windows (version 24.0), as well as Microsoft Excel 2016 (Microsoft Corp, Redmond, WA, USA). A *p*-value ≤ 0.050 denoted statistical significance.

## 3. Results

Frequencies and percentages of adherence to the MD are shown in Figure 1. No statistically significant differences were found by sex, showing a similar pattern between boys and girls. Overall, 8.9% of the schoolchildren had very low diet quality (*n* = 33), while 65.1% needed improvement to adjust intake to Mediterranean patterns (*n* = 241). Furthermore, only 25.9% of the schoolchildren had optimal adherence to the MD (*n* = 96).

Data on age, anthropometric characteristics, test of PF, KIDMED index score and PA level of the sample according to adherence to the MD are shown in Table 1. Regarding the scores of the different PF tests in these three groups, the only statistically significant differences were found for CRF (*p* = 0.048; *d* = 0.29) in boys. Subsequently, post-hoc comparisons tests only showed these differences between Low MD and High MD. In the case of PA level, high MD group showed higher scores in PAQ-C, in both sexes. Apart from that, statistically significant differences were observed in both boys (*p* = 0.040; *d* = 0.30) and girls (*p* = 0.016; *d* = 0.40). However, post-hoc comparisons tests only showed significant differences between low MD and both moderate and high MD in boys; being between moderate MD and high MD for girls.

Table 2 shows the different correlations observed according to the KIDMED index score and the different variables as age, BMI, PF and PA. A positive statistically significant correlation was found between the KIDMED index score and PA level (*p* = 0.235). Notwithstanding, all these correlations were low.

Finally, Table 3 indicates the probability of obtaining different results in the continuous variables of the study according to the category of adherence to the MD. Thus, it is observed a higher probability of having a greater CRF (OR = 1.17; CI = 1.02–1.34) and PA level (OR = 7.84; CI = 2.84–21.60), when schoolchildren are classified as High MD.

## 4. Discussion

This research attempted to describe, compare and analyse the level of PF and PA in schoolchildren aged 6–13 in the Region of Murcia, according to sex and adherence to the MD. The main results of this study suggest that optimal adherence to the MD seems to be associated to a greater CRF, especially among boys. It was also associated with a higher level of PA for both boys and girls. Similarly, participants classified as High MD were more likely to express higher CRF, as well as a higher level of PA, when they were compared to participants with classified as Low and Moderate MD.

MD has justified its efficacy in the primary and secondary prevention of cardiovascular diseases, with the highest level of scientific evidence [43]. In addition to this, it has recently been demonstrated that better adherence to the MD is associated with a better lipid profile and adiposity measures and, in the case of women, with the reduction of some cardiovascular risk factors [44]. In spite of these enormous health benefits, among the participants of the current study, it is observed that only one out of four presents an optimal adherence to the MD. This low prevalence coincides with the data reported by other authors [26], and it is possible to glimpse the trend indicated in the scientific literature towards leaving of the Mediterranean pattern [18,19].

In relation to sex, no differences between boys and girls were found, as it has been reported in other studies [22,45]. In contrast, with regards to the PF test, as occurred in a large number of studies, boys scored higher than girls; as well as for the case of PA level. This disparity in results according to sex may be related to the different moments and levels of evolutionary development, since these may determine a greater performance of the males in the PF tests [46].

Regarding anthropometric measurements, no statistically differences were found based on BMI, WC or BF in both boys and girls. This fact agrees with a recent systematic review conducted by Iaccarino et al. [26], which indicated that most studies performed found no statistically significant differences for weight status, abdominal obesity and adiposity in relation to MD adherence. However, it should be underlined that exploring the relationship between MD and overweight/obesity is complicated, since KIDMED index does not include the frequency of consumption of certain foods, as well as the estimated nutritional intakes derived from this [47].

In order to increase primary prevention of cardiovascular risk, apart from the relevance of consuming foods characteristic of the MD, it must not be forgotten that adherence to this pattern also leads to an active lifestyle [13]. In this line, we found a positive relationship between adherence to MD and PA. This fact is in line with the findings of Shi et al. [48], who found that those children who take healthy meals are more likely to be physically active and less likely to be sedentary when compared with their counterparts who consume less healthy meals; concluding that having healthy eating habits is an indicator for adopting a more active lifestyle. This relationship has been studied in Mediterranean children [49] and different determinants, such as mother’s education or screen time could influence the existing relationship. In addition, other consequences of having low level of PA where observed in children with lower adherence to MD such as higher frequency of sedentary behaviours, increased blood pressure or higher BMI. Respect to the explanation for this relationship, previous studies principally carried out in adolescents, consider that individuals that practice more PA could want a better performance and they choose better food to achieve it [50] or the fact that more PA requires more energy expenditure and individuals reporting more PA would have higher intakes of essential nutrients [51].

On the other hand, we observed that a greater adherence to the MD is associated with a greater CRF, as also demonstrated other studies [18,23,24]. This fact could be explained to some degree by the above-mentioned relationship with PA, since a higher participation in PA (as a result of active lifestyle) could increase PF level [52]. Likewise, recent studies have showed an association between muscle strength and optimal adherence to the MD [19,20,21]. Thus, the combination of an optimal adherence to MD, followed by appropriate levels of muscular fitness, seems to provide the highest protection against cardiometabolic risk [21]. Apart from that, it has been found that optimal adherence to the MD could not be enough to prevent some deleterious health effects if adequate levels of muscle strength are not available [24]. These relationships have also been found in the current sample, although it was not statistically significant. Notwithstanding, contrary to the studies mentioned above, no data on different biochemical parameters (glucose, cholesterol, etc.) were analysed in our study.

This study presented certain limitations that must be explained. First, due to the cross-sectional design of the present study, it is not be able to conclude that the observed relationships reflect causal relationships. Secondly, although the KIDMED index is the instrument most commonly used to determine the adherence to the MD, it may have been interesting to obtain information on the frequency of consumption of certain foods characteristic of the Mediterranean pattern. On the other hand, although we used validated PA questionnaires, we did not use accelerometer devices, which would have provided a more accurate assessment of PA level and sedentary behaviour. Finally, the fact of not having considered the state of development of the participants could have introduced a confusing effect.

## 5. Conclusions

These results suggest that an optimal adherence to the MD is associated with higher CRF and PA level in schoolchildren aged 6–13 years in the Region of Murcia (Spain) who participated in the present study. Moreover, it was found that only 25.9% of schoolchildren that took part in the study presented an optimal adherence to MD. Authors point out that awareness-raising campaigns are also needed to warn of the adverse health effects of inappropriate eating habits and low levels of PF. At the same time, more prospective cohort studies are required that could clarify better the relationships among adherence to the MD, PF and PA, as well as with other health-related parameters.

## Figures and Tables

**Figure 1 nutrients-12-00567-f001:**
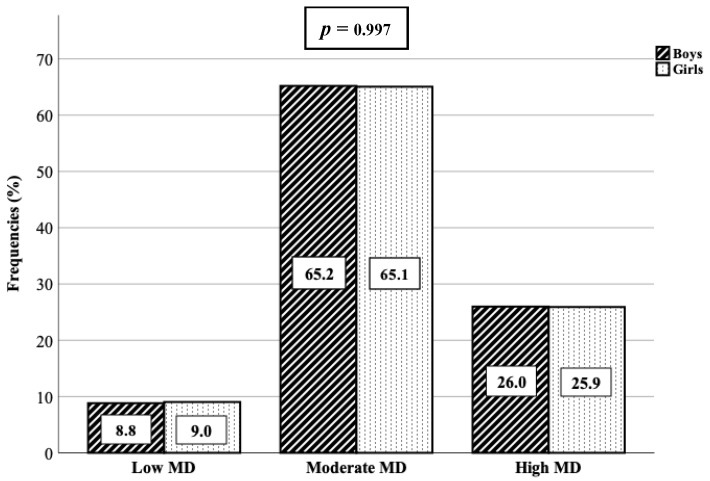
Adherence to the Mediterranean Diet in schoolchildren aged 6–13 in the Region of Murcia (Spain).

**Table 1 nutrients-12-00567-t001:** Sample descriptive data according to adherence to the Mediterranean Diet.

Variables	Boys					Girls				
	Low MD *n* = 18 (8.8%)	Moderate MD *n* = 133 (65.2%)	High MD *n* = 53 (26.0%)	*p*	*d*	Low MD *n* = 15 (9.0%)	Moderate MD *n* = 108 (65.1%)	High MD *n* = 43 (25.9%)	*p*	*d*
Age (years)	9.1 ± 1.3	8.9 ± 1.8	8.5 ± 1.9	0.378	0.03	8.7 ± 1.7	8.4 ± 1.8	8.6 ± 1.8	0.779	0.19
Weight (kg)	39.07 ± 10.77	36.19 ± 10.15	36.03 ± 12.23	0.451	0.09	33.30 ± 12.05	34.04 ± 10.34	37.48 ± 12.38	0.208	0.17
Height (m)	1.39 ± 0.07	1.37 ± 0.11	1.36 ± 0.13	0.637	0.15	1.34 ± 0.12	1.34 ± 0.12	1.35 ± 0.14	0.812	0.20
BMI (kg/m^2^)	20.07 ± 4.23	18.96 ± 3.39	19.02 ± 3.68	0.560	0.13	18.11 ± 3.84	18.73 ± 3.69	20.07 ± 4.41	0.043*	0.33
BMI (*z*-score)	0.84 ± 0.94	1.12 ± 1.36	1.27 ± 1.04	0.385	0.04	0.99 ± 0.88	1.20 ± 1.25	1.09 ± 1.17	0.832	0.20
WC (cm)	64.27±5.98	63.18 ± 7.52	62.33 ± 8.6	0.202	0.16	59.18 ± 8.13	60.35 ± 8.88	62.64 ± 8.03	0.120	0.24
WHtR (WC/Height (cm))	0.46 ± 0.04	0.46 ± 0.04	0.46 ± 0.04	0.884	0.19	0.44 ± 0.04	0.45 ± 0.06	0.46 ± 0.05	0.082	0.27
BF (kg)	12.50 ± 6.39	10.91 ± 5.43	11.04 ± 6.46	0.491	0.11	10.72 ± 6.23	10.77 ± 5.78	12.76 ± 6.73	0.161	0.20
BF (%)	30.40 ± 7.68	28.76 ± 6.36	28.67 ± 7.06	0.632	0.15	30.14 ± 7.12	29.88 ± 7.70	32.35 ± 7.40	0.181	0.19
FFM (kg)	26.56 ± 4.83	25.29 ± 5.21	24.99 ± 6.12	0.509	0.11	22.58 ± 6.04	23.27 ± 5.26	24.72 ± 6.41	0.328	0.08
FFM (%)	69.60 ± 7.68	71.24 ± 6.36	71.33 ± 7.06	0.632	0.15	69.86 ± 7.12	70.12 ± 7.70	67.65 ± 7.40	0.181	0.19
Handgrip strength (kg)	13.08 ± 2.46	13.86 ± 4.07	13.12 ± 4.04	0.433	0.18	12.51 ± 4.35	11.91 ± 3.54	12.90 ± 4.19	0.339	0.23
Handgrip strength/BW	0.34 ± 0.08	0.39 ± 0.08	0.38 ± 0.09	0.154	0.27	0.38 ± 0.05	0.36 ± 0.08	0.35 ± 0.07	0.484	0.19
Standing broad jump (cm)	120.2 ± 29.9	116.2 ± 24.6	120.5 ± 26.9	0.538	0.16	108.6 ± 25.8	106.6 ± 22.2	111.2 ± 23.5	0.537	0.18
4 × 10 m Shuttle Run Test (s)	13.49 ± 1.44	13.52 ± 1.64	13.57 ± 1.25	0.710	0.16	14.32 ± 1.53	14.05 ± 1.11	14.03 ± 1.61	0.690	0.18
20 m Shuttle Run Test (laps)	16.1 ± 12.8 ^a^	21.0 ± 14.3	23.4 ± 13.4	0.048 *	0.29	15.5 ± 7.1	16.3 ± 9.7	17.9 ± 9.6	0.412	0.08
CRF (mL/kg/min)	43.31 ± 4.00 ^a^	45.20 ± 4.29	46.36 ± 4.56	0.019 *	0.35	43.98 ± 4.03	44.60 ± 4.01	44.89 ± 3.73	0.699	0.18
PAQ-C (score)	1.87 ± 0.53 ^a,b^	2.23 ± 0.45	2.28 ± 0.52	0.040 *	0.30	1.96 ± 0.38	2.04 ± 0.40 ^a^	2.22 ± 0.45	0.016 *	0.40

Data expressed as Mean ± SD. BMI: Body mass index; BW: Body weight; CRF: Cardiorespiratory Fitness; BF: Body fat; FFM: Free-fat mass; MD: Mediterranean Diet; WC: Waist circumference; WHtR: Waist-to-Hip ratio. ^a^ Significantly different from High MD *(p ≤* 0.050). ^b^ Significantly different from Moderate MD *(p ≤* 0.050). * *p ≤* 0.050.

**Table 2 nutrients-12-00567-t002:** Bivariate and partial correlations between KIDMED index score and different variables of the study.

Variables	Crude	Adjusted ^#^
Handgrip strength (kg)	−0.015 (0.779)	0.065 (0.213)
Handgrip strength/BW	−0.010 (0.841)	0.040 (0.442)
Standing broad jump (cm)	0.069 (0.188)	0.147 ** (0.005)
4 x 10 m Shuttle Run Test (s)	0.002 (0.967)	−0.057 (0.276)
20 m Shuttle Run Test (laps)	0.104 * (0.045)	0.141 ** (0.007)
CRF (mL/kg/min)	0.134 ** (0.010)	0.144 ** (0.006)
PAQ-C (score)	0.162 ** (0.002)	0.232 *** (<0.001)

BW: Body weight; CRF: Cardiorespiratory Fitness. * *p ≤* 0.050; ** *p ≤* 0.010; *** *p ≤* 0.001. ^#^ Adjusted by sex, age and BMI *z*-score, WC and %BF.

**Table 3 nutrients-12-00567-t003:** Analysis of multinomial logistic regression for continuous variables of study according to the Mediterranean Diet.

Predictors	Model 1			Model 2		
	Low MD (0–3)	Moderate MD (4–7)	High MD (≥8)	Low MD (0–3)	Moderate MD (4–7)	High MD (≥8)
Handgrip strength (kg)	1	1.01 (0.92–1.11)	1.01 (0.92–1.12)	1	1.09 (0.93–1.26)	1.11 (0.94–1.32)
Handgrip strength/BW	1	6.68 (0.08–576.95)	1.68 (0.01–212.40)	1	11.30 (0.05–2648.77)	9.52 (0.03–3625.12)
Standing broad jump (cm)	1	1.00 (0.98–1.01)	1.00 (0.99–1.02)	1	0.99 (0.97–1.01)	1.01 (0.99–1.03)
4 × 10 m Shuttle Run Test (s)	1	0.95 (0.74–1.22)	0.96 (0.73–1.25)	1	0.91 (0.67–1.25)	0.83 (0.59–1.17)
20 m Shuttle Run Test (nº laps)	1	1.03 (0.99–1.06)	1.04 * (1.00–1.08)	1	1.03 (0.99–1.07)	1.06 ** (1.01–1.11)
CRF (mL/kg/min)	1	1.08 (0.99–1.18)	1.13 * (1.02–1.25)	1	1.11 (0.96–1.28)	1.23 ** (1.06–1.44)
PAQ-C (score)	1	3.72 ** (1.49–9.28)	6.24 *** (2.33–16.70)	1	4.06 ** (1.58–10.44)	7.51 *** (2.71–20.86)

BW: Body Weight; CRF: Cardiorespiratory Fitness; MD: Mediterranean Diet. * *p ≤* 0.050; ** *p ≤* 0.010; *** *p ≤* 0.001. Model 1: Unadjusted; Model 2: Adjusted by sex, age, BMI *z*-score, WC and %BF.

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
