# Peer review of "Adherence to Mediterranean Diet Related with Physical Fitness and Physical Activity in Schoolchildren Aged 6–13"

_nutrients, 2020, doi:10.3390/nu12020567_

Round 1

Reviewer 1 Report

Interesting article about children's health. Important due to childhood obesity, which is one of the major concerns in the past years and is mainly due to the consumption of unhealthy foods and reduced levels of physical activity.

Below are presented detailed comments, which according to the reviewer, should be included to the manuscript or should be responded to by the Authors.

Abstract: the authors used abbreviations (CRF and KIDMED). They should be explained

Materials and methods:

-how were completed the questionnaires? by the children themselves? with the help of parents? Please complete this information

-“A total of 370 schoolchildren (204 boys 80 and 166 girls) aged 6–13 participated in the study” – and the PAQ-C is recommended for children between 8–14 years old. Please explain this choice

- Currently, the ALFA-FIT test components are shown as the following subsections and as other research methods. I suggest changing the section numbering format (2.2.2.1. Anthropometric measurements etc.)

-  Statistical Analysis - The reviewer wonders about the correctness of the use of the Mann-Whitney U test in the case of post hoc?

Results:

-The authors showed  the correlation “A positive statistically significant correlation was found between the KIDMED index score and CRF (ρ=.127) and standing broad jump (ρ=.133).”  however a correlation below 0.2 means no relationship

-table 1 “a” and “b” in notes have the same subtitles (a Significantly different from High MD (p≤.050). b Significantly different from High MD (p≤.050)).

The comments presented of the manuscript do not diminish its substantive value.

I recommend publishing the manuscript after prior correction of the text or the Authors’ reference to the specific comments.

Author Response

Thanks for your valuable comments. We agree and we consider too important to describe the situation in different regions with specific features due to we have already global and country information about this matter.

RESPONSE TO REVIEWERS

REVIEWER 1

Interesting article about children's health. Important due to childhood obesity, which is one of the major concerns in the past years and is mainly due to the consumption of unhealthy foods and reduced levels of physical activity.

Thanks for your valuable comments. We agree and we consider too important to describe the situation in different regions with specific features due to we have already global and country information about this matter.

Below are presented detailed comments, which according to the reviewer, should be included to the manuscript or should be responded to by the Authors.

We have tried to address all issues mentioned and we have answered each of them.

Abstract: the authors used abbreviations (CRF and KIDMED). They should be explained

Thanks for the comment. CRF was already described in the abstract at the first appearance. However, we have explained KIDMED, and PAQ-C that has been added.

Materials and methods:

-how were completed the questionnaires? by the children themselves? with the help of parents? Please complete this information

We have added new information in the methods section as we show below.

a)

Materials and Methods:

Both the PF tests and the filling-out of the questionnaires were carried out by the schoolchildren during the physical education sessions.

b)

2.2.3. Physical activity:

PAQ-C was validated for children aged 8-14. For this reason, in the case of children aged 6-7, the indications of Bervoets et al. [42] were followed, as it recommended that parents should be encouraged to support their children with reading and to complete the questions, provided that they did not give any guidance in answering the questions.

  1. Bervoets, L.; Van Noten, C.; Van Roosbroeck, S.; Hansen, D.; Van Hoorenbeeck, K.; Verheyen, E.; Van Hal, G.; Vankerckhoven, V. Reliability and Validity of the Dutch Physical Activity Questionnaires for Children (PAQ-C) and Adolescents (PAQ-A). Arch. Public Health 2014, 72, 47.

-“A total of 370 schoolchildren (204 boys 80 and 166 girls) aged 6–13 participated in the study” – and the PAQ-C is recommended for children between 8–14 years old. Please explain this choice

In addition to a previous answer, we consider that PAQ-C has been validated in different languages and used with children aged 5 to 17, so we have followed the same criteria because it has been demonstrated that PAQ-C provides reliable and valid estimates of PA in this age range. This is of great clinical utility since it makes it possible to design personalised interventions to prevent and combat obesity as well as the pathologies associated with it (Bervoets et al. 2014).

- Currently, the ALFA-FIT test components are shown as the following subsections and as other research methods. I suggest changing the section numbering format (2.2.2.1. Anthropometric measurements etc.)

Yes, we consider that it is better, thanks for the contribution. We have changed the division of the different tests for a better understanding. If it is not clear, please do not hesitate to indicate to us.

-  Statistical Analysis - The reviewer wonders about the correctness of the use of the Mann-Whitney U test in the case of post hoc?

We have clarified this part of the text as follows: “with Bonferroni correction to account for the inflation of type-I error due to multiple comparisons made”.

Results:

-The authors showed the correlation “A positive statistically significant correlation was found between the KIDMED index score and CRF (ρ=.127) and standing broad jump (ρ=.133).”  however a correlation below 0.2 means no relationship

Thanks for the contribution because we consider that could create confusion, so we have deleted it.

-table 1 “a” and “b” in notes have the same subtitles (a Significantly different from High MD (p≤.050). b Significantly different from High MD (p≤.050)).

Thanks for the observation. It was a mistake and we have amended it as follows: “Significantly different from Moderate MD”.

The comments presented of the manuscript do not diminish its substantive value.

I recommend publishing the manuscript after prior correction of the text or the Authors’ reference to the specific comments.

Thanks for your comments and your valuable labour reviewing our manuscript. We hope to have addressed all the specific comments you have provided us.

Reviewer 2 Report

Overall:

The aim of this research was to describe, compare and analyze the level of PF and PA in schoolchildren aged 6-13 in 20 the Region of Murcia, according to adherence to the MD. The authors writing is very clear and easy to follow. Moreover, they have done a very good job of organizing a lot of data and making it easy to interpret. The reviewer has minor recommendations for the authors.

Abstract:

The methods discussion should further discuss the PF tests. They are listed in results, but prior to that it is unknown what PF tests the students engaged in.

Line 27: Need to state in methods that KDMED index is used to determine adherence to MD.  

Introduction:

Line 43: delete On the other hand

Line 43: should be rewritten to say there are several factors that influence PF, such as……

Line 48: delete on this matter

Line 52: the younger ones? Does this mean children?

Line 54: delete we must not forget that both

Line 68: Other than type of food variances between South America and Spain, why might you speculate differences in adherence? Is there some other underlying factor that would explain why this is an important research question to address?

Methods:

Overall this section is written very well.

Line 162: should say activity they engaged in.

Results:

This section is also written very well. The authors are commended for their work.

Discussion

Line 28: The authors should discuss this issue in more detail and how this should be evaluated in further research.

Line 48-49: because of cross-sectional design that the present study, should be reworded.

Author Response

We would like to thank your comments. We are happy seeing that we have achieved our aim of doing it clear for the reader. We have tried to address all the comments provided in the review.

REVIEWER 2

Overall:

The aim of this research was to describe, compare and analyze the level of PF and PA in schoolchildren aged 6-13 in 20 the Region of Murcia, according to adherence to the MD. The authors writing is very clear and easy to follow. Moreover, they have done a very good job of organizing a lot of data and making it easy to interpret. The reviewer has minor recommendations for the authors.

We would like to thank your comments. We are happy seeing that we have achieved our aim of doing it clear for the reader.

We have tried to address all the comments provided in the review.

Abstract:

The methods discussion should further discuss the PF tests. They are listed in results, but prior to that it is unknown what PF tests the students engaged in.

We tried to reduce the text but probably we deleted relevant information. Then, we agree with your comment and we have added new information to do it clearer:   Mediterranean Diet Quality Index for children and teenagers (KIDMED) was used to determinate the adherence to the MD. ALPHA-FIT Test Battery was applied for assess body composition and PF. PA level was determined using Physical Activity Questionnaire for Older Children (PAQ-C).

Line 27: Need to state in methods that KDMED index is used to determine adherence to MD. 

With the addition of new information it has been already included in the previous comment.

Introduction:

Line 43: delete On the other hand

We have deleted it.

Line 43: should be rewritten to say there are several factors that influence PF, such as……

We have deleted the first part of the sentence and we consider that now it explains the meaning better.

Line 48: delete on this matter

After reading it again, we agree because the previous paragraph explains different factors. So, we have changed the connector in order to do it clearer for the reader.

Line 52: the younger ones? Does this mean children?

Yes, it is. However, we changed to “children” in order to clarify this aspect.

Line 54: delete we must not forget that both

We have deleted it.

Line 68: Other than type of food variances between South America and Spain, why might you speculate differences in adherence? Is there some other underlying factor that would explain why this is an important research question to address?

We have reviewed carefully the text mentioned and what we would like to say is that even with healthy eating habits in different regions around the world, the KIDMED score could vary because of differences in foods. So, we think that we have not speculated about adherence but we should point out the influence of differences in food on the diet assessment. So, we have added information to clarify that influence:

Moreover, the differences among continents on food environment, defined as all aspects of the local environment that influence diet (i.e. community, organizational and consumer food environment), could have influence on the eating patterns [28].

  1. Franco, M.; Bilal, U.; Díez, J. Food Environment. In Encyclopedia of Food and Health, Caballero, B., Finglas, P. M., Toldrá, F., Eds.; Elsevier, 2016; pp. 22–26.

Methods:

Overall this section is written very well.

Line 162: should say activity they engaged in.

We have changed it.

Results:

This section is also written very well. The authors are commended for their work.

Thanks for the comment. It is important for us in order to continue in this way as always trying to improve our skills.

Discussion

Line 28: The authors should discuss this issue in more detail and how this should be evaluated in further research.

As we have reported in the introduction, there are few studies on this topic focused in children. Grosso and Galvano [49] did a review on adherence to Mediterranean diet in four countries at Mediterranean region. An association between MD and PA was found in all of the countries analysed (Spain, Greece, Cyprus and Italy) although not all the participants were children. So, we considered not discussing it deeply because of the previous evidence. However, after considering your contribution, we have added more information as follows:

This relationship has been studied in Mediterranean children [49] and different determinants, such as mother’s education or screen time could influence the existing relationship. In addition, other consequences of having low level of PA where observed in children with lower adherence to MD such as higher frequency of sedentary behaviours, increased blood pressure or higher BMI. Respect to the explanation for this relationship, previous studies principally carried out in adolescents, consider that individuals that practice more PA could want a better performance and they choose better food to achieve it [50] or the fact that more PA requires more energy expenditure and individuals reporting more PA would have higher intakes of essential nutrients [51].

  1. Grosso, G.; Galvano, F. Mediterranean diet adherence in children and adolescents in southern European countries. NFS J. 2016, 3, 13–19.
  2. Zurita-Ortega, F.; San Román-Mata, S.; Chacón-Cuberos, R.; Castro-Sánchez, M.; Muros, J. Adherence to the Mediterranean Diet Is Associated with Physical Activity, Self-Concept and Sociodemographic Factors in University Student. Nutrients 2018, 10, 966.
  3. Chacón-Cuberos, R.; Zurita-Ortega, F.; Martínez-Martínez, A.; Olmedo-Moreno, E.; Castro-Sánchez, M. Adherence to the Mediterranean Diet Is Related to Healthy Habits, Learning Processes, and Academic Achievement in Adolescents: A Cross-Sectional Study. Nutrients 2018, 10, 1566.

Line 48-49: because of cross-sectional design that the present study, should be reworded.

We have modified it as follows: “due to the cross-sectional design of the present study”.

Reviewer 3 Report

The paper entitled "Adherence to Mediterranean Diet related with physical fitness and physical activity in schoolchildren aged 6-13" raises an important topic, as the influence of a very healthy eating pattern as the Mediterranean Diet on children physical condition and performance. As psychologist, I do not feel competent for judging the measures employed in the study, but as methodologist I can say that the data analyses are clear and logical. The discussion part is well structured and covers the issues presented in the introduction and in the methodological section. 

Author Response

We would like to thank your comments. We are happy seeing that we have achieved our aim of doing it clear and logical for the reader.